# SHARED CONTEXTS, PERSONALIZED OUTPUTS: A BENCHMARK FOR DOCUMENT GENERATION

## ABSTRACT

Large Language Models (LLMs) have recently demonstrated strong capabilities in long-context text generation, enabling applications such as meeting summarization and multi-document question answering. However, these tasks typically focus on producing a single, context-consistent output, without accounting for user-specific roles, preferences, or intents. *Personalized Contextual Document Generation* (PCDG) requires models to generate distinct, user-tailored documents grounded in the same extended context. Generating user-tailored outputs is key to adaptive applications, reducing manual edits and improving downstream utility, yet this capability remains underexplored due to the difficulty of evaluation. Furthermore, benchmarking PCDG effectively demands realistic, controllable context modeling, explicit personalization signals, well-defined intermediate sub-tasks, and evaluation metrics that go beyond surface-level similarity. To this end, we present **PersonaContextWeaver**, a benchmarking framework designed to meet these requirements through three key innovations: (1) a knowledge-graph-based synthesis pipeline that generates rich, multi-user, cross-domain conversational contexts with controllable personalization variables; (2) a task decomposition strategy that evaluates not only final document quality but also intermediate reasoning steps including as intent detection, context filtering, reference prediction; and (3) a multi-dimensional evaluation protocol to evaluate LLM's capability of user intent/profile understanding, relevant context retrieval, and document customization. Empirical evaluation of state-of-the-art LLMs on **PersonaContextWeaver** reveals substantial gaps in their ability to consistently generate highly personalized, contextually accurate documents. Models often struggle with nuanced user modeling, context filtering, and reference integration, indicating that personalized contextual document generation remains a challenging frontier for current LLMs.

## 1 INTRODUCTION

Large language models (LLMs) have achieved remarkable progress in long-context text generation Liu et al. (2024); Yang et al. (2025). Models such as GPT-4, Claude, and Gemini can process inputs containing tens of thousands of tokens, enabling applications like meeting summarization Laskar et al. (2023), multi-document question answering Wang et al. (2024), and code synthesis over large repositories Nijkamp et al.. These advances demonstrate that LLMs can maintain coherence, track entities, and integrate information across extended sequences. Recently, increasing efforts have catalyzed a paradigm shift in content generation, moving from generic, one-size-fits-all generation to Personalized Generation Xu et al. (2025).

Personalized contextual document generation (PCDG) introduces an additional layer of complexity beyond traditional long-context generation. In PCDG, the objective is not only to produce a coherent document grounded in an extended input context, but also to **tailor the output to a specific user's role, preferences, and intent**. For example, given the transcript of a cross-functional two-hour product design meeting, a project manager might require a risk-focused project update, while a marketing lead might request a customer-facing product announcement. Both outputs must remain factually consistent with the same underlying conversation, yet differ in focus, tone, and information prioritization. Such personalization not only improves communication relevance but also reduces time spent editing generic outputs and enables systems to learn user-specific priorities over time and ultimately enhancing productivity and decision-making. Achieving this level of personalization requires nuanced context filtering, selective content emphasis, and stylistic adaptation. These capabilities are extended beyond the scope of existing long-context generation benchmarks, necessitating a dedicated benchmark framework to systematically evaluate them.

Building a benchmark for PCDG introduces unique requirements. Unlike generic summarization or dialogue synthesis tasks, PCDG evaluation must account for multiple dimensions simultaneously. Specifically, a comprehensive benchmark should: (1) provide realistic and controllable contextual grounding so that personalization is well-defined

Table 1: Comparison of benchmarks across personalization, context modeling, reasoning depth, evaluation framework, persona-awareness, cross-context reasoning, and document generation. ✓: strong support; △: partial support; ✗: none.

| Attributes / Benchmark | Personalization | Context Modeling | Long Reasoning | Evaluation Framework | Persona-based | Cross-Context | Document Generation |
|---|---|---|---|---|---|---|---|
| Gehrmann et al. (2021) | △ | ✗ | △ | △ | ✗ | ✗ | △ |
| Yin et al. (2021) | ✗ | ✗ | △ | △ | ✗ | ✗ | ✗ |
| Kumar et al. (2024) | ✓ | △ | △ | ✓ | ✗ | ✗ | ✓ |
| Wu et al. (2025) | ✗ | ✓ | ✓ | ✓ | ✗ | ✗ | ✓ |
| Tan et al. (2025) | ✓ | ✗ | ✗ | ✓ | ✓ | ✗ | △ |
| Chang et al. (2024) | △ | ✓ | ✓ | ✓ | ✓ | ✗ | ✓ |
| Li et al. (2025) | △ | ✓ | ✓ | ✓ | ✗ | ✗ | ✓ |
| Lin (2025) | ✗ | ✓ | ✓ | ✓ | ✗ | ✓ | ✗ |
| Peper et al. (2025) | ✗ | ✓ | ✓ | ✓ | ✗ | ✓ | ✗ |
| **PersonaContextWeaver** | ✓ | ✓ | ✓ | ✓ | ✓ | ✓ | ✓ |

and reproducible, (2) formally define the problem space and intermediate subtasks such as intent detection, context filtering, and intent–document alignment, and (3) establish evaluation methodology that capture personalization quality and contextual fidelity beyond surface-level similarity to reference document. Without these components, it is difficult to systematically assess progress or compare approaches.

Several benchmarks have advanced research in areas such as long-context reasoning, dialogue generation, and persona-based interaction (Table 1). For instance, LongGenBench Wu et al. (2025) and LongLaMP Kumar et al. (2024) evaluate models' ability to handle extended contexts and maintain coherence, while PersonaBench Tan et al. (2025) focuses on persona-driven dialogue. Similarly, AgentBoard Chang et al. (2024) and CaseGen Li et al. (2025) explore structured workflows and domain-specific document generation. These benchmarks provide valuable insights into individual aspects such as long-context reasoning, personalization, or task-specific generation. However, none are explicitly designed to address the combined requirements of PCDG; namely, generating multiple personalized documents from a shared, multi-user context while ensuring contextual fidelity, intent alignment, and role-awareness. This motivates our development of a dedicated benchmark that integrates these dimensions into a unified benchmark framework.

Designing such a benchmark is nontrivial, it requires creating rich, multi-user conversations that span multiple topics and phases, with personalization signals clearly represented to enable systematic evaluation. Three main challenges arise. First, real-world multi-user conversations are extremely difficult to obtain because they often contain sensitive or private information. Second, existing publicly available conversational datasets are not suitable for PCDG. Most open-domain datasets lack structured continuity and task-oriented context. For instance, a Reddit thread on "UI design tips" may contain scattered opinions without a clear timeline or actionable decisions, making it impossible to generate a coherent status report or progress summary. Finally, generating reliable gold-standard personalized outputs requires precise control over both the shared context and personalization dimensions (e.g., ensuring the context supports a risk-focused summary and the output reflects the intended tone and audience). Without such control, fair and effective evaluation becomes very challenging.

To address these challenges, we introduce **PersonaContextWeaver**, a benchmark suite for Personalized Contextual Document Generation. The benchmark begins by constructing synthetic conversational contexts using a graph-driven generation pipeline, designed to capture realistic multi-user, multi-topic, and multi-phase interactions while preserving traceability of contextual elements. Building on these contexts, we define a structured evaluation pipeline with intermediate tasks: personalized intent detection, context filtering, and document generation. Each task is assessed through a combination of automated metrics, LLM-as-judge evaluations, and context-grounding checks, enabling fine-grained analysis of both reasoning steps and final outputs. Our main contributions are summarized as follows:

- **Benchmark Design:** We propose the first benchmark tailored for PCDG, featuring a graph-driven synthesis pipeline that produces realistic, multi-user, cross-domain conversational contexts enriched with persona diversity, temporal structure, and implicit personalization cues.

- **Task Decomposition:** We formalize PCDG into interpretable sub-tasks including personalized intent detection, context filtering, and document generation. This enables fine-grained evaluation of intermediate reasoning steps.

- **Evaluation Framework:** We introduce a multi-dimensional evaluation protocol to evaluate LLM's capability of user intent/profile understanding, relevant context retrieval, and document customization.

- **Empirical Study:** We benchmark state-of-the-art LLMs on PersonaContextWeaver, providing insights into current limitations and potential directions for personalized generation.

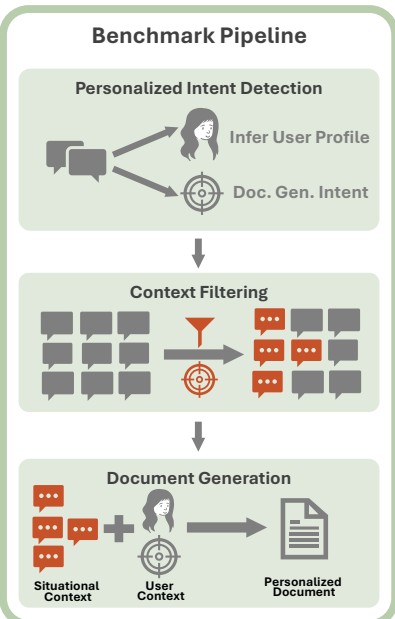
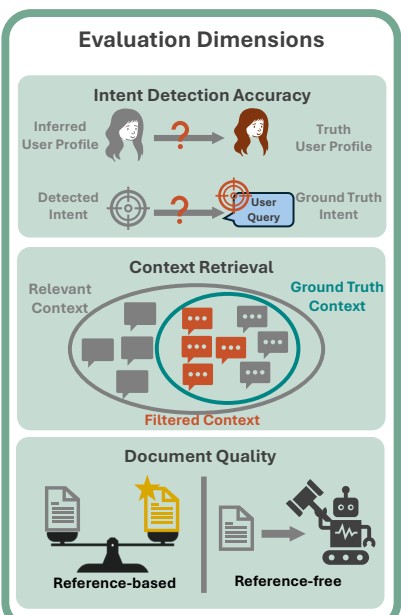

Figure 1: **Overview of PersonaContextWeaver.** The benchmark consists of three components: (1) *Context and User Query:* Synthetic multi-user conversations are generated with role-specific attributes and temporal structure, paired with user queries that require reasoning over distributed context and personalization cues. (2) *Benchmark Pipeline:* The evaluation process decomposes document generation into subtask: Personalized Intent Detection, Context Filtering, and Document Generation—ensuring traceability and interpretability of intermediate reasoning steps. (3) *Evaluation Dimensions:* Models are assessed on intent detection accuracy, context retrieval quality, and document quality using both reference-based metrics and reference-free methods.

## 2 CONTEXTUAL PERSONALIZATION: DEFINITION AND GROUNDING

In this section, we first provide an overview of the benchmark by formally define the problem of contextual personalized document generation and introduce the design and structure of the PersonaContextWeaver. Then, we illustrate on the methodology of synthetic data generation.

### 2.1 BENCHMARK OVERVIEW

**Problem Formulation.** We define the task of contextual personalized document generation (PCDG) as producing textual outputs that are not only relevant to a given user query but also tailored to the user's situational context. This context includes role, intent, domain expertise, and interaction history within a shared multi-user environment. Let $U = \{u_1, \ldots, u_N\}$ be a set of users, each associated with a contextual profile $\mathcal{C}_u$ that captures structured and unstructured signals such as prior authored content, project affiliations, domain-specific expertise, temporal markers (e.g., deadlines, recency), and communication style. For each generation instance, we define the situational context $\mathcal{S}_u$ as the set of contextual elements (e.g., historical messages, metadata, and relevant interactions) retrieved from the user's history and recent activity that are relevant to the user's current objective.

Given a dataset $\mathcal{T} = \{(\mathcal{Q}_1, \mathcal{D}_1, \mathcal{C}_{u_1}), \ldots, (\mathcal{Q}_M, \mathcal{D}_M, \mathcal{C}_{u_M})\}$, the goal is to learn a conditional generation function that maps a prompt $\mathcal{Q}$, a user context $\mathcal{C}_u$, and a set of situational context $\mathcal{S}_u$ to a document $\hat{\mathcal{D}}$: $\hat{\mathcal{D}} \sim p(\cdot \mid \mathcal{Q}, \mathcal{C}_u, \mathcal{S}_u)$, where the goal is for $\hat{\mathcal{D}}$ to closely approximate the ground truth $\mathcal{D}$. This task requires reasoning over the semantic content of $\mathcal{Q}$, the user-specific signals in $\mathcal{C}_u$, and the situational grounding provided by $\mathcal{S}_u$. The generated document must satisfy the following criteria:

- **Personalization**: Reflect the user's role, goals, and stylistic preferences;
- **Contextual consistency**: Align with factual, temporal, and domain-specific constraints present in $\mathcal{C}_u$ and $\mathcal{S}_u$;
- **Coherence and relevance**: Form a logically connected whole and remains on-topic with respect to $\mathcal{Q}$.

This formulation enables benchmarking models on their ability to generalize across users, domains, and temporal settings, while leveraging both persistent user profiles and dynamically retrieved situational context.

Table 2: Expanded relation schema in the conversation knowledge graph, including semantic meaning and cardinality.

| Relation | Source $\rightarrow$ Target | Type | Semantic Meaning | Cardinality |
|---|---|---|---|---|
| belongs_to_domain | $\tau \rightarrow d$ | Structural | Topic is part of a domain | Many-to-one |
| has_topic | $d \rightarrow \tau$ | Structural | Domain contains a topic | One-to-many |
| has_phase | $\tau \rightarrow \phi$ | Structural | Topic unfolds across phases | One-to-many |
| belongs_to_topic | $\phi \rightarrow \tau$ | Structural | Phase is associated with a topic | Many-to-one |
| related_to_topic | $\tau_i \leftrightarrow \tau_j$ | Semantic | Topics are conceptually related | Many-to-many |
| involved_in | $u \rightarrow \tau$ | Social | User participates in a topic | Many-to-many |
| has_member | $\tau \rightarrow u$ | Social | Topic includes a user | One-to-many |
| active_in_phase | $u \rightarrow \phi$ | Social | User is active during a phase | Many-to-many |
| has_phase_member | $\phi \rightarrow u$ | Social | Phase includes a user | One-to-many |
| author_of | $u \rightarrow m$ | Interactional | User authored a message | One-to-many |
| authored_by | $m \rightarrow u$ | Interactional | Message was written by a user | One-to-one |
| discussed_in_topic | $m \rightarrow \tau$ | Semantic | Message contributes to a topic | Many-to-one |
| discussed_in_phase | $m \rightarrow \phi$ | Semantic | Message is part of a phase | Many-to-one |
| replies_to | $m_i \rightarrow m_j$ | Interactional | Message replies to another message | One-to-one |

**Benchmark Design and Structure.** As illustrated in Figure 1, **PersonaContextWeaver** is organized into three interconnected components that mirror the panels in the diagram:

**(1) Context and User Query.** Each benchmark instance is constructed through a controlled data synthesis pipeline rather than relying on raw historical logs. We generate a *situational context* that simulates realistic multi-user conversations across domains, topics, and project phases. This context is created using a graph-driven generation process during benchmark construction, but models only receive the resulting conversation text, not the underlying graph. The synthetic conversation embeds role-specific behaviors, communication styles, and domain expertise to reflect authentic collaboration dynamics.

Alongside the conversation, we generate a *user query* that specifies the document generation goal (e.g., "Create a status report for the planning phase covering progress and next steps"). Queries are designed to require reasoning over multiple turns and personalization signals distributed throughout the context. This preparation ensures that each instance captures the complexity of real-world scenarios including multi-user interactions, evolving topics, and implicit personalization cues, while maintaining full control over ground truth for evaluation.

**(2) Benchmark Pipeline.** The pipeline decomposes PCDG into three subtasks:

- *Personalized Intent Detection:* From the query and situational context, the model infers a structured intent schema (document type, audience, tone, temporal scope) and approximates user profile cues.
- *Context Filtering:* The model selects relevant messages and metadata from the shared context, forming an evidence set for grounding document generation.
- *Document Generation:* Using the inferred intent and filtered context, the model produces a personalized document aligned with user goals and situational constraints.

**(3) Evaluation Dimensions.** PersonaContextWeaver evaluates models along multiple perspectives:

- *Intent Detection Accuracy:* Comparing predicted intent schema and inferred profile against ground truths.
- *Context Retrieval:* Measuring precision, recall of selected message IDs relative to the ground truth evidence set.
- *Document Quality:* Evaluating document quality using both reference-based metrics (BLEU, ROUGE) with a semi-automatically curated set of golden documents, and reference-free methods (i.e., LLM-as-judge) to assess readability, redundancy, personalization fidelity, and temporal accuracy.

This design ensures that models are assessed not only on final output quality but also on intermediate reasoning steps critical for contextual personalization.

## 2.2 REALISTIC SYNTHETIC CONVERSATION

**Grounding Structure.** We model the user's conversational context as a directed knowledge graph $\mathcal{G} = (\mathcal{V}, \mathcal{E})$, where $\mathcal{V}$ includes *domains* ($d$), *topics* ($\tau$), *phases* ($\phi$), *users* ($u$), and *messages* ($m$), and $\mathcal{E}$ encodes semantic relations among these entities (see Table 2). This graph-based representation encodes the structured, relational aspects of the user's contextual profile $\mathcal{C}_u$, capturing both semantic and temporal dependencies across user interactions. By grounding generation in $\mathcal{G}$, models can reason over user-specific trajectories, domain constraints, and discourse structure—supporting both personalization and contextual consistency in the generated document.

**Synthesizing Context Structure.**

To generate a new message node, our framework traverses the knowledge graph $\mathcal{G}$ using three predefined path types, each capturing a distinct conversational scenario:

- **Context Initialization** (domain → topic → phase → user): establishes the initial conversational context by selecting a relevant user within a specific domain, topic, and phase.

- **Local Interaction** (message → user → phase → user): captures intra-phase interactions, such as replies or follow-ups between users engaged in the same discussion thread.

- **Context Transition** (message → topic → topic → phase → user): enables transitions across related topics and phases, supporting continuity in cross-team or cross-domain conversations.

Each traversal yields a coherent set of contextual nodes, including domain, topic, phase, and target user. These contextual nodes jointly define the conditions under which the new message is generated. Based on these conditions, we retrieve relevant historical messages and metadata to form the situational context $\mathcal{S}_u$ for the current generation instance. This graph-guided approach enables the model to incorporate information from multiple threads and phases, supporting realistic and coherent multi-turn dialogue continuity.

For example, given a sampled **Local Interaction Path** (message $(m)$ → phase $(\phi)$ → user $(u)$), the objective is to generate a synthetic message for the target user $u$ in response to the original message $m$, within the same phase $\phi$. This path captures intra-phase interactions, such as a team member replying to a colleague's update during the planning phase of a project. To achieve this, we retrieve messages associated with the phase $(m \to \phi)$ to construct the situational context $\mathcal{S}_u$, and condition the generation on both $\mathcal{S}_u$ and the user profile $\mathcal{C}_u$. The generated message node $m \in \mathcal{V}$ is instantiated with core metadata $(u, t)$, indicating its author $u$ and timestamp $t$. Once created, the node is linked to its corresponding user $u$, phase $\phi$, and the original message node $m$. This structured representation ensures that synthetic messages remain coherent with the ongoing conversation and accurately reflect interactions spanning multiple discussion threads and project phases.

**Prompting Language Model for Message Generation.**

Given the synthesized context structure from graph traversal in $\mathcal{G}$, including the target user $u$, relevant domain, topic, phase nodes, and associated historical messages, the message generation process leverages prompt engineering, using the situational context $\mathcal{S}_u$ and user profile $\mathcal{C}_u$, to produce realistic, personalized messages with a large language model. Specifically, for each message to be generated, we construct a prompt that encodes:

- The user's persona attributes (e.g., role, tone, style, expertise) from $\mathcal{C}_u$
- The current project, topic, and phase
- The situational context $\mathcal{S}_u$ (retrieved messages and metadata relevant to the current instance)
- The intended conversational scenario (e.g., reply, discussion, cross-role interaction)

Depending on the path type (e.g., initial, discussion, role_role), the prompt is dynamically adapted to instruct the LLM to generate a message that fits the user's style and the ongoing thread. The example prompts can be found in Appendix C. For reply scenarios, the prompt includes explicit mention of the previous author and message, ensuring continuity and coherence. Then, the constructed prompt is passed to the LLM (e.g., GPT-5), which generates a candidate message $\hat{m}$:

$$\hat{m} \sim p(\cdot \mid \text{prompt}(\mathcal{S}_u, \mathcal{C}_u)).$$

The generated message is instantiated as a node in $\mathcal{G}$, annotated with metadata (author, timestamp, path type), and linked to relevant entities (user, project, phase, reply-to message).

**Quality Control and Integration.** To ensure realism and diversity, the framework occasionally introduces noise messages—plausible but subtly incorrect or off-topic responses—to simulate human error. Each generated message is evaluated for coherence, personalization, and structural consistency before being integrated into the evolving conversational graph.

Table 3: Conversational domains and their attributes

| Dataset | # Domains / Topics / Phases | # Participants | # Roles | # Posts | # Replies | Avg. token per message |
|---|---|---|---|---|---|---|
| Technology | 7 / 32 / 172 | 19 | 6 | 947 | 3,553 | 145.7 |
| Healthcare | 10 / 48 / 247 | 22 | 5 | 1,004 | 3,496 | 138.9 |
| Manufacturing | 10 / 48 / 246 | 18 | 5 | 979 | 3,530 | 143.2 |
| Finance | 10 / 49 / 239 | 24 | 9 | 977 | 3,523 | 140.6 |

This approach enables scalable synthesis of multi-turn, contextually grounded, and personalized team conversations, supporting downstream benchmarking and analysis of LLM-based dialogue agents. Table 3 list the a set of generated group chat messages covers across 4 domains with two types of message: post and reply.

We evaluate the synthesized data by extending G-Eval Liu et al. (2023) to group chat scenario with latest large language model (i.e., GPT-5). The evaluation metrics including: Naturalness, Coherence, Diversity, Contextual Relevance, Momentum, Engagingness. Each metrics scores from 0 to 5, the higher the better. To show the quality of synthetic dataset, we also apply these metrics to real-world dataset and a baseline synthetic dataset that leverage single prompt to generate group messages. The evaluation is run across 10 different random seeds with same temperature. More details can be found in Appendix C.5

Table 4: Synthetic Data Quality Evaluation with LLM-as-Judge of G-Eval Liu et al. (2023)

| Dataset | Naturalness | Coherence | Diversity | Contextual Relevance | Momentum | Engagingness | Overall Avg. |
|---|---|---|---|---|---|---|---|
| Baseline (Single Prompt) | 4.00 ± 0.00 | 4.00 ± 0.00 | 3.70 ± 0.30 | 4.00 ± 0.00 | 4.30 ± 0.46 | 3.10 ± 0.49 | 3.85 |
| Upper Bound (Real-world) | 5.00 ± 0.00 | 4.90 ± 0.30 | 5.00 ± 0.00 | 5.00 ± 0.00 | 5.00 ± 0.00 | 5.00 ± 0.00 | 4.98 |
| Technology | 5.00 ± 0.00 | 4.60 ± 0.49 | 5.00 ± 0.00 | 5.00 ± 0.00 | 5.00 ± 0.00 | 5.00 ± 0.00 | 4.93 |
| Healthcare | 5.00 ± 0.00 | 4.90 ± 0.30 | 5.00 ± 0.00 | 5.00 ± 0.00 | 5.00 ± 0.00 | 5.00 ± 0.00 | 4.98 |
| Manufacturing | 5.00 ± 0.00 | 5.00 ± 0.00 | 4.70 ± 0.46 | 5.00 ± 0.00 | 5.00 ± 0.00 | 4.90 ± 0.30 | 4.93 |
| Finance | 5.00 ± 0.00 | 4.60 ± 0.49 | 4.90 ± 0.30 | 5.00 ± 0.00 | 5.00 ± 0.00 | 4.90 ± 0.30 | 4.90 |

## 2.3 SYNTHETIC USER QUERY GENERATION

To simulate realistic user requests for document generation, we define three types of target documents: *Status Report*, *Email*, and *FAQ Document*. Each document type corresponds to a specific communicative intent and is grounded in high-level context extracted from the knowledge graph $\mathcal{G} = (\mathcal{V}, \mathcal{E})$, which encodes semantic and structural relationships among domains, topics, phases, users, and messages.

Unlike message generation, which relies on graph traversal to simulate conversational flow, query synthesis focuses on summarizing the conversation to extract salient signals that a user would naturally reference when requesting a document. These signals—referred to as *contextual markers*—are derived from the content of message nodes and include discussed topics, completed phases, key decisions, unresolved questions, and user roles.

To extract these markers, we perform lightweight summarization over the message nodes connected to the target node (e.g., phase $\phi$ for status reports, topic $\tau$ for emails). This involves:

- Identifying frequently mentioned entities (e.g., topics, tasks, tools).
- Detecting temporal expressions and phase-specific updates.
- Extracting user actions such as requests, decisions, or clarifications.
- Aggregating message-level metadata (e.g., authorship, timestamps).

The **synthesis module** then combines these contextual markers with user persona features $\mathcal{C}_u$ to generate a concise, natural language query. This module can be implemented as a rule-based template engine, a retrieval-augmented prompt constructor, or a lightweight LLM-based generator fine-tuned for query synthesis. The output is a one-sentence request that reflects the user's intent and situational context, such as:

**Status Report**: "Summarize our recent progress and upcoming focus areas for Incident Response and Recovery on MonitoringAgent."

**Email**: "Provide a brief update on predictive maintenance progress, recent changes, and resource allocation for the Downtime Reduction Taskforce."

**FAQ Document**: "Update on the team's training and engagement efforts for the patient experience project, including major developments and current challenges."

The query serves as input to the document generation pipeline, ensuring that the output is contextually grounded, semantically aligned, and tailored to realistic user behavior. Note that we intentionally make the document generation intent implicit in user queries to better benchmark the model's intent detection capability. In total, we synthesized 160 user queries for document generation, Table 5 is the summary of the queries.

Table 5: Summary statistics of the 160 synthesized user queries for document generation.

| Dataset | Num. Queries | Status Reports | Emails | FAQs | Avg. Query Length | Avg. Num. Labels |
|---|---|---|---|---|---|---|
| Technology | 40 | 17 | 15 | 8 | 56.2 | 71.2 |
| Healthcare | 40 | 15 | 17 | 8 | 60.6 | 47.5 |
| Manufacturing | 40 | 19 | 14 | 7 | 56.0 | 49.1 |
| Finance | 40 | 17 | 16 | 7 | 54.6 | 45.6 |

In addition to generating the synthetic query, we also extract the corresponding ground-truth label for each instance. Specifically, for every query, we collect the set of relevant message nodes from the knowledge graph that are temporally available up to the query timestamp and contextually linked to the target phase or topic. These message IDs serve as the gold standard for context retrieval and evaluation, enabling precise benchmarking of model performance. The full

details of our label extraction process including marker types, temporal filtering, and output structure, are provided in Appendix C.6.

# 3 BENCHMARK TASKS & EVALUATION

## 3.1 PERSONALIZED INTENT DETECTION

**Task Definition.** Given a document generation query $\mathcal{Q}$ and a situational context $\mathcal{S}_u$, which includes relevant conversational history, associated metadata, the personalized intent detection task requires the model to infer two components: (i) a structured intent schema $\hat{\mathcal{I}}$ derived primarily from $\mathcal{Q}$, capturing fields such as *Document Type*, *Target Audience*, *Focus Area*, *Tone/Style*, and *Temporal Scope*; and (ii) an inferred user profile summary $\hat{\mathcal{C}}_u$ reconstructed from $\mathcal{S}_u$, approximating the contextual profile $\mathcal{C}_u$ defined in our problem formulation (e.g., *Role*, *Domain Expertise*, *Preferences*, *Communication Style*). This design reflects real-world scenarios where persistent user profiles are unavailable, and personalization must emerge from context. Formally:

$$(\mathcal{Q}, \mathcal{S}_u) \mapsto \{\hat{\mathcal{I}}, \hat{\mathcal{C}}_u\}.$$

The inferred $\hat{\mathcal{C}}_u$ serves as a proxy for $\mathcal{C}_u$ in downstream document generation, ensuring that outputs are aligned with user-specific goals and contextual signals.

**Evaluation Metrics.** The predicted schema $\hat{\mathcal{I}}$ and inferred profile $\hat{\mathcal{C}}_u$ are compared against gold-standard annotations $\mathcal{I}^*$ and $\mathcal{C}_u^*$. We compute:

- **User Profile Accuracy**: Semantic similarity between inferred and reference profile fields.
- **Intent Capture Accuracy**: Strict exact match for categorical fields. Detailed per-field metrics for intent schema is provided in Appendix D.

We report per-field accuracy, mean semantic similarity, and overall F1, enabling fine-grained evaluation of both intent interpretation and inferred personalization.

## 3.2 CONTEXT FILTERING

**Task Definition.** Given an inferred user intent schema $\hat{\mathcal{I}}$ and a situational context $\mathcal{S}_u$ consisting of multi-user message history and associated metadata, the context filtering task requires the model to identify all relevant messages that serve as the knowledge source for document generation. Each message is represented by a unique identifier (message_id), and the goal is to produce a subset $\hat{\mathcal{R}} \subseteq \mathcal{S}_u$ such that $\hat{\mathcal{R}}$ contains all and only those messages necessary to satisfy the user's intent while preserving contextual fidelity. Formally:

$$(\hat{\mathcal{I}}, \mathcal{S}_u) \mapsto \hat{\mathcal{R}} = \{\text{message\_id}_1, \dots, \text{message\_id}_k\}.$$

This step operationalizes the grounding process by ensuring that downstream document generation is supported by explicit, traceable evidence from the shared context.

**Evaluation Metrics.** We compare the predicted set $\hat{\mathcal{R}}$ against the gold-standard reference set $\mathcal{R}^*$ using:

- **Precision**: Proportion of retrieved messages that are relevant.
- **Recall**: Proportion of relevant messages that are retrieved.
- **F1-score**: Harmonic mean of precision and recall.

This evaluation ensures that models not only retrieve relevant evidence but also maintain correct attribution for factual grounding in the final output.

## 3.3 DOCUMENT GENERATION

**Task Definition.** Given the inferred intent schema $\hat{\mathcal{I}}$, the filtered evidence set $\hat{\mathcal{R}}$, and the situational context $\mathcal{S}_u$, the document generation task requires the model to produce a personalized document $\hat{\mathcal{D}}$ that satisfies three criteria: (1) **Personalization** — aligns with the inferred intent schema (e.g., tone, audience, temporal scope); (2) **Contextual Fidelity** — grounds all factual content in the retrieved evidence set; (3) **Coherence and Completeness** — forms a logically structured, fluent document that addresses the user query. Formally:

$$(\hat{\mathcal{I}}, \hat{\mathcal{R}}, \mathcal{S}_u) \mapsto \hat{\mathcal{D}}.$$

**Evaluation Methods.**   Document generation is evaluated along two complementary ways:

- **Reference-based Quality Evaluation:** Compare generated outputs against curated golden documents using BLEU, ROUGE, and TextGrad-based similarity.
- **Reference-free Quality Evaluation:** Assess personalization fidelity, fluency, and factual consistency without relying on a single reference.

### 3.3.1   REFERENCE-FREE QUALITY EVALUATION

Reference-based metrics often fail to capture personalization quality or factual grounding when multiple valid outputs exist. To address this, we adopt *reference-free evaluation* using LLM-as-judge and task-specific scoring rubrics.

**Evaluation Metrics.**   Each generated document is assessed along the following dimensions:

- **Citation Accuracy (0.0–1.0):** Proportion of evidence-linked claims that include correct message IDs from $\hat{\mathcal{R}}$.
- **Personalization Fidelity (1–5):** Degree to which the document reflects the inferred intent schema (tone, audience, temporal scope).
- **Factuality (1–5):** Accuracy of content relative to $\hat{\mathcal{R}}$, verified via LLM-based consistency checks.
- **Fluency (1–5):** Grammaticality and readability of the text.
- **Structure (1–5):** Logical organization and adherence to expected section layout.
- **Temporal Accuracy (1–5):** Alignment of temporal expressions with the specified project phase or time frame.

We implement this using GPT-4.1-based evaluators with structured prompts and 1–5 scoring scales for each dimension. Final scores are aggregated into a weighted composite metric, enabling robust evaluation even when multiple correct outputs exist.

### 3.3.2   REFERENCE-BASED QUALITY EVALUATION

Reference-based evaluation measures how closely a generated document matches a curated golden reference. This approach ensures comparability across systems while capturing surface-level and semantic quality.

**Golden Document Preparation.**   For each synthetic query, we construct a high-quality reference document through an iterative workflow inspired by TextGrad Yuksekgonul et al. (2024):

- Generate an initial draft using our best-performing model, grounded in the query context and intent.
- Chunk the draft into logical sections for fine-grained evaluation.
- Evaluate each section using LLM-as-judge metrics (e.g., accuracy, structure, personalization) to identify issues.
- Revise low-scoring or problematic sections through targeted LLM editing or manual curation.
- Repeat evaluation and editing until all sections meet quality thresholds.

This TextGrad-inspired, iterative process ensures that each golden document is both contextually accurate and personalized. Full details are provided in Appendix C.7.

**Evaluation Metrics.**   Generated documents are compared against golden references using:

- **ROUGE (ROUGE-1, ROUGE-L):** Measures lexical overlap by computing n-gram and longest common subsequence matches between the generated output and the reference.
- **METEOR:** Captures both exact and semantically similar matches (including stemming and synonyms), providing a more nuanced assessment of content similarity.

## 4   BENCHMARK RESULTS

We evaluate five OpenAI models—GPT-4.1 OpenAI (2025a), GPT-4o OpenAI (2024), O4-mini OpenAI (2025c), GPT-5-chat OpenAI, and GPT-5 OpenAI (2025b) on the PersonaContextWeaver benchmark. Table 6 summarizes their performance across three core tasks: profile & intent detection, context filtering, and reference-free document generation. Full experiment setting details, examples, and prompt templates are provided in Appendix D. Based on the result, we make following observations.

First, all models perform comparably on user profile and intent detection. GPT-4.1 achieves the highest user profile accuracy (0.50), while GPT-4o leads in intent accuracy (0.53). These results align with OpenAI's positioning of GPT-4.1 as a model optimized for instruction following and long-context comprehension OpenAI (2025a), and GPT-4o as a

Table 6: **Performance Metrics for PersonaContextWeaver Tasks.**

| Task | Metric | GPT-4o | GPT-4.1 | O4-mini | GPT-5-chat | GPT-5 |
|------|--------|--------|---------|---------|-----------|-------|
| **Profile & Intent Detection** | User Profile Accuracy ↑ | 0.48 | 0.50 | 0.45 | 0.45 | 0.48 |
| | Intent Accuracy ↑ | 0.53 | 0.50 | 0.49 | 0.50 | 0.46 |
| **Context Filtering** | Precision ↑ | 0.27 | 0.19 | 0.16 | 0.21 | 0.25 |
| | Recall ↑ | 0.14 | 0.16 | 0.11 | 0.19 | 0.25 |
| | F1-score ↑ | 0.16 | 0.17 | 0.12 | 0.19 | 0.25 |
| **Document Generation (Reference-Free)** | Citation Accuracy (0.0 - 1.0) ↑ | 0.10 | 0.13 | 0.12 | 0.17 | 0.22 |
| | Personalization Fidelity (1-5) ↑ | 4.39 | 4.13 | 4.42 | 4.74 | 4.19 |
| | Factuality (1-5) ↑ | 4.31 | 3.68 | 4.20 | 4.95 | 3.48 |
| | Fluency (1-5) ↑ | 4.99 | 5.00 | 4.93 | 4.97 | 4.93 |
| | Structure (1-5) ↑ | 4.50 | 4.45 | 4.89 | 4.91 | 4.72 |
| | Temporal Accuracy (1-5) ↑ | 4.29 | 4.19 | 4.65 | 4.89 | 4.18 |
| **Document Generation (Reference-based)** | ROUGE-1 ↑ | 0.35 | 0.37 | 0.28 | 0.31 | 0.41 |
| | ROUGE-L ↑ | 0.13 | 0.13 | 0.11 | 0.12 | 0.13 |
| | METEOR ↑ | 0.22 | 0.22 | 0.13 | 0.18 | 0.24 |

fast, general-purpose model designed for multimodal and real-time interaction OpenAI (2024). More detailed analysis is provided in Appendix D.5

Second, for context filtering, GPT-5 achieves the highest F1-score (0.25), with balanced precision and recall, followed closely by GPT-5-chat. This reflects GPT-5's architectural emphasis on structured reasoning and long-context retrieval. GPT-4o also performs well in precision (0.27), consistent with its design for efficient inference. O4-mini, while lightweight, lags slightly in recall, which is expected given its optimization for latency and cost rather than deep context modeling.

Third, GPT-5-chat stands out with the highest scores in personalization fidelity (4.74), factuality (4.95), structure (4.91), and temporal accuracy (4.89), showing its strength in producing coherent, user-aligned outputs. GPT-5 achieves the best citation accuracy (0.22), suggesting strong evidence retrieval capabilities. However, its lower factuality score (3.48) indicates a disconnect between citation and semantic integration. This may reflect a trade-off that the model excels at identifying relevant context but does not consistently integrate that information faithfully into the generated content. GPT-4.1 and O4-mini remain strong in fluency and structure, while GPT-4o delivers competitive fluency (4.99) but lower citation grounding (0.10), consistent with its focus on speed and multimodal interaction.

Forth, turning to the reference-based evaluation, we observe that GPT-5 achieves the highest ROUGE-1 and METEOR scores, indicating strong lexical and semantic overlap with the golden references, while other models show more modest gains. However, these results present a notable contradiction when compared to the reference-free evaluation. That is, GPT-5-chat excels in personalization fidelity, factuality, and structure according to LLM-as-judge metrics, but it does not lead in reference-based scores. This divergence highlights a key challenge in personalized document generation that reference-based metrics may penalize valid stylistic and structural diversity, especially when multiple correct outputs exist, whereas reference-free metrics are more sensitive to personalization and contextual alignment. By capturing and discussing this discrepancy, our benchmark demonstrates a nuanced understanding of the evaluation landscape and underscores the necessity of multi-dimensional assessment.

To summarize, GPT-5-chat is the most capable model for generating personalized, contextually grounded documents. GPT-5 shows promise in context filtering and citation grounding but may require further refinement to improve factual consistency. GPT-4.1 and GPT-4o remain reliable for user modeling and fluency, while O4-mini offers a cost-effective baseline with solid performance across tasks. These findings underscore the importance of aligning model selection with the specific personalization and grounding requirements of downstream applications.

## 5 CONCLUSION

We present **PersonaContextWeaver**, a comprehensive benchmark for Personalized Contextual Document Generation (PCDG).We enables fine-grained evaluation of both intermediate reasoning and final output quality by synthesizing realistic multi-user conversational contexts and decomposing the document generation process into interpretable subtasks. Empirical results across state-of-the-art LLMs reveal that, while recent models excel in fluency and personalization, substantial challenges remain in context filtering, factual grounding, and nuanced user modeling. These findings underscore the need for continued research into context-aware personalization and robust evaluation protocols. We hope PersonaContextWeaver will serve a new standard for evaluating personalized document generation and paves the way for future progress in this field.

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

# Shared Contexts, Personalized Outputs: A Benchmark for Document Generation

## Supplemental Material

## A    REPRODUCIBILITY

To ensure full reproducibility of our benchmark and results, we provide all code, scripts, and benchmark outputs as part of the supplemental material. These resources are packaged in a zip file accompanying the submission.

The archive includes:

- Source code and scripts for running all benchmark experiments
- Generated datasets and evaluation pipelines
- Example configuration files and usage instructions
- Complete benchmark results and summary statistics
- All details and artifacts described in the Appendix sections, including prompts, evaluation protocols, and metric definitions

Every step of the benchmark pipeline, from user profile and intent schema detection, context retrieval, document generation, to reference-free LLM judging, are documented and reproducible using the provided materials. The dataset, synthetic queries, and evaluation outputs are included to enable independent verification and extension of our results.

We intend to open source the full benchmark suite after completion of our internal privacy and compliance review. This will ensure that all materials meet organizational standards for responsible data sharing.

## B    THE USE OF LARGE LANGUAGE MODELS

Large Language Models (LLMs) play a multifaceted role in both the development and evaluation of our benchmark and in the preparation of this paper.

**Paper Preparation**    LLMs were used extensively for proof-reading and grammatical checking throughout the writing process. All sections of the manuscript were reviewed using LLMs to ensure clarity, correctness, and consistency in language.

**Benchmark Dataset and Synthetic Query Generation**    For the benchmark itself, LLMs were employed to generate synthetic user queries and conversations. This enabled the creation of diverse, realistic scenarios for evaluating document generation and context retrieval capabilities.

**Reference-free Judging**    A core innovation of our benchmark is the use of LLMs as reference-free judges. For each generated document, an LLM is prompted to evaluate quality across multiple dimensions (e.g., personalization fidelity, factuality, citation quality, fluency, structure, temporal accuracy), producing detailed metric scores and feedback in a fully automated manner.

**Benchmark Design and Implementation**    Importantly, the overall design, orchestration, and implementation of the benchmark pipeline—including all code and scripts for running experiments, collecting results, and managing evaluation were developed by the authors with limited LLM assistance such as debugging and docstring writing. The logic for data loading, intent schema extraction, context retrieval, and document generation orchestration is entirely author-written.

**Statistical Summarization**    For post-processing and summarization of benchmark results, we occasionally leveraged LLMs to generate scripts for statistical analysis and visualization. This use was limited to auxiliary tasks and did not affect the core benchmark logic or evaluation methodology.

**Summary** In summary, LLMs were used for (1) proof-reading and grammar checking of the paper, (2) generating synthetic data and queries, (3) automated judging of benchmark outputs, and (4) assisting with statistical summarization scripts. The benchmark's conceptual design and implementation remain author-driven, ensuring methodological transparency and reproducibility.

## C  PROMPT ENGINEERING FOR MESSAGE GENERATION, SYNTHETIC USER QUERIES, AND LABEL GENERATION

### C.1  PROMPT CONSTRUCTION LOGIC

Message generation in PersonaContextWeaver is grounded in the knowledge graph $\mathcal{G}$, which encodes domains, topics, phases, users, and messages, along with their semantic and temporal relations. For each new message node, the framework traverses $\mathcal{G}$ using predefined path types—**Context Initialization**, **Local Interaction**, and **Context Transition**—to synthesize realistic, multi-user conversational scenarios. Prompts are constructed to encode:

- **Persona attributes:** Role, tone, style, expertise.
- **Project, topic, and phase:** Current context for the message.
- **Situational context** $\mathcal{S}_u$: Recent messages, metadata, and relevant history.
- **Conversational scenario:** Initial post, reply, cross-role/cross-project interaction, or noise injection.

Depending on the graph traversal path, the prompt is tailored to fit the user's style and the ongoing thread, ensuring continuity and authenticity.

### C.2  PATH-SPECIFIC PROMPT TEMPLATES AND EXAMPLES

**Context Initialization (Channel Post)**  *Scenario:* A user posts an announcement or kickoff message at the start of a phase.

**Prompt Template:**

> You're {user}, creating a new post in your Teams channel about the {phase_name} phase of {project}.
> Your role: {role}
> Your expertise: {expertise}
> Your style: {tone} and {style}
> Current status: {status}
> Target date: {target_date}
> Write a channel post that announces this phase and gets the team engaged.
> - Make it clear what this phase is about and why it matters
> - Include key information the team needs to know
> - Ask questions or request input to start discussion
> - Use your natural voice and style
> - Keep it professional but approachable
> - Think about what would make people want to reply and contribute

**Example Output:**

> Welcome to the Planning phase, team! Our goal is to outline project risks and set clear milestones. Please review the objectives and share any initial concerns or ideas. Looking forward to a productive start!

**Local Interaction (Threaded Reply)**  *Scenario:* A user replies to a colleague's message within the same phase.

**Prompt Template:**

> Background: {user} is a {role} with expertise in {expertise}, communicating in a {tone} tone and {style} style.
> Phase: {phase_name} ({stage} stage – {progress}% complete)

Status Goal: {status}
Target Date: {target_date}
Channel Conversation History: {recent_messages}
Task: Write a REPLY in the {project} channel thread. Respond to: {reply_to}
Style Guide:
- Reply in a casual, conversational style
- Build on what others have said
- Use your {tone} tone and {style} style
- Keep it concise (1–2 sentences)
- Use @mentions and emojis if appropriate
Content Focus:
- Respond to blockers or questions
- Share updates from your area
- Propose solutions or alternatives

**Example Output:**

@*Sam* Good catch on the authentication issue! I'll sync with the backend team and see if we can patch it by tomorrow.

**Context Transition (Cross-Context / Role-to-Role)**   *Scenario:* A user from one project/phase joins a discussion in another project's channel, sharing insights or collaborating across domains.

**Prompt Template:**

Background: {user} is a {role} with expertise in {expertise}, communicating in a {tone} tone and {style} style.
They are currently working on the {source_project} project, in the {source_phase} phase.
They are joining a cross-project, same-role discussion in the {target_project} channel, in a thread about the {topic} domain during its {target_phase} phase.
Recipient is also a {role}.
Phase Context: {target_phase} ({stage} stage – {progress}% complete)
Status Goal: {status}
Original Post: {post_content}
Thread Context: {recent_messages}
Task:  Write a REPLY in this Teams channel thread that reflects your experience from {source_project}.
Guidance:
- Share insights from your {source_project} experience that apply here
- Ask thoughtful questions or suggest approaches based on what you've learned
- Highlight overlaps or synergies between {source_project} and {target_project}
- Keep it conversational and helpful—you're a peer offering perspective
- Use your {tone} tone and {style} style
- End with a question or suggestion that moves the discussion forward

**Example Output:**

Henry, in Healthcare Analytics we found that combining qualitative interviews with quantitative risk scoring helped surface hidden issues. Have you considered integrating stakeholder feedback into your risk models? Curious if your team has faced resistance to new methodologies.

**Noise Message**   *Scenario:* Simulates plausible but subtly incorrect or off-topic messages for realism.

**Prompt Template:**

{user} is a {role} who communicates in a {tone} tone and {style} style.
They are discussing the topic-{topic}, posting a message during the {phase_name} phase of the {project} project.
Owner: {owner}, Status: {status}, Target Date: {target_date}
Thread Context: {recent_messages}

> Write a plausible but subtly incorrect, confused, or off-topic message that a real team member might send in this context.
> Examples: misunderstanding the next step, referencing the wrong deadline, asking an unrelated but believable question, or making a human error.
> The message should still sound natural and fit the ongoing conversation.

**Example Output:**

> Wait, are we supposed to finish the UI by next Monday or was it the API? Sorry, I got mixed up with the deadlines.

### C.3 PROMPT ADAPTATION AND QUALITY CONTROL

Prompts are dynamically adapted based on:

- **Role and expertise:** Ensures messages reflect authentic perspectives.
- **Phase progress:** Early-stage prompts focus on planning; late-stage prompts drive closure.
- **Communication purpose:** Kickoff, update, blocker, decision, milestone, escalation, coordination.
- **Conversation history:** Maintains continuity and realism in threaded discussions.
- **Noise injection:** Introduces human-like errors and off-topic remarks for realism.

Each generated message is evaluated for coherence, personalization, and structural consistency before being integrated into the evolving conversational graph.

### C.4 CONTROLLING CONVERSATION PROGRESSION

A key aspect of PersonaContextWeaver is the dynamic control of conversation progression, ensuring that synthetic dialogues unfold in a realistic, temporally coherent manner across phases, topics, and projects. The progression logic is tightly coupled to the knowledge graph $\mathcal{G}$ and leverages both graph traversal and message history to simulate organic team interactions.

**Phase and Message Selection.** For each new message, the system selects a phase and topic by traversing $\mathcal{G}$ using relation paths (e.g., has_topic, has_phase), with optional filters to ensure phases are active and have not yet been completed. The selection process is probabilistic and balances between domains, topics, and phases to avoid repetitive or stagnant conversations.

**Post and Reply Scheduling.**

- **Channel Posts:** The type of post (e.g., kickoff, milestone, update, blocker, decision) is determined by the number of existing messages in the phase and the current stage of progression. Early posts tend to be announcements or kickoffs, while later posts reflect ongoing work, blockers, or decisions.
- **Replies:** Replies are scheduled by selecting recent messages within a phase, prioritizing those that have not received sufficient responses. The system avoids repeatedly selecting the same authors, fostering diverse participation.
- **Cross-Project Replies:** For cross-context scenarios, the system identifies users with matching roles across related projects and phases, enabling knowledge transfer and collaborative problem-solving between teams.

**Temporal Realism.** Each message is assigned a timestamp using a bursty, Poisson-like process, ensuring that conversations reflect natural activity patterns (e.g., bursts of replies, lulls between phases). The system checks for phase completion before scheduling new messages, preventing unrealistic activity in closed phases.

**Progression Guidance in Prompts.** Prompts are dynamically adapted to the current stage of the phase (early, middle, late), with explicit guidance for the LLM to focus on appropriate actions:

- **Early Stage:** Emphasize planning, problem identification, and team alignment.
- **Middle Stage:** Focus on solution development, status updates, and collaborative progress.

- **Late Stage:** Drive closure, confirm deliverables, and resolve outstanding issues.

For example, a late-stage prompt may include: "We're at 90% through this phase (ending soon). The phase MUST achieve 'Completed' status. Focus on finalizing decisions, confirming completions, and closing out open items."

**Quality Control.** The system evaluates each generated message for coherence, relevance, and structural consistency before integrating it into the evolving conversation. Noise messages are occasionally injected to simulate human error and maintain realism.

## C.5 SYNTHETIC CONVERSATION QUALITY

**Synthetic Conversation Evaluation Metrics** For all synthetic group chat data, we report quality using six core metrics, each scored by LLM-as-Judge protocols (e.g., G-Eval) on a scale from 1 to 5, with higher values indicating better performance:

- **Naturalness:** Measures how authentic and human-like the conversation appears. High scores indicate messages resemble real workplace communication, with natural phrasing and plausible imperfections.

- **Coherence:** Assesses the logical flow and connectivity between messages. Coherent conversations maintain topic continuity and clear transitions, avoiding abrupt or confusing shifts.

- **Diversity:** Captures the variety in communication styles, topics, and participant contributions. High diversity reflects a mix of message lengths, tones, and perspectives, avoiding repetitive exchanges.

- **Contextual Relevance:** Evaluates how well each message relates to the shared context and ongoing discussion. Relevant conversations consistently reference prior messages, project details, and team objectives.

- **Momentum:** Measures the degree to which the conversation progresses toward goals or resolutions. High momentum indicates that messages drive the discussion forward, address blockers, and facilitate decision-making.

- **Engagingness:** Assesses how interactive and stimulating the conversation is for participants. Engaging conversations feature active participation, questions, acknowledgments, and responses that encourage further dialogue.

- **Overall Avg.:** The arithmetic mean of the above metrics, providing a single summary score for overall conversation quality.

These metrics collectively capture the realism, effectiveness, and collaborative dynamics of synthetic workplace conversations, enabling nuanced evaluation and comparison across datasets.

**Analysis of Synthetic Conversation Quality Metrics** Table 4 presents a comparative evaluation of synthetic group chat data across six key metrics, benchmarked against both a baseline (single prompt) and real-world conversations. First, the real-world upper bound achieves near-perfect scores ($\geq 4.9$) across all metrics, confirming that authentic workplace conversations are consistently natural, coherent, diverse, contextually relevant, goal-driven, and engaging. Second, the baseline (single prompt) scenario shows moderate performance, with overall average quality (3.85) notably lower than real-world data. While naturalness, coherence, and contextual relevance are reasonably high (4.00), diversity (3.70) and engagingness (3.10) are limited, suggesting that single-prompt generation tends to produce repetitive and less interactive exchanges. Third, synthetic datasets generated by PersonaContextWeaver (Technology, Healthcare, Manufacturing, Finance) approach real-world quality, with overall averages between 4.90 and 4.98. Naturalness, contextual relevance, momentum, and engagingness consistently reach the maximum score (5.00 or close), indicating that the synthesis pipeline effectively captures the authentic dynamics of workplace group chats. Fourth, minor variations are observed in coherence and diversity across domains. For example, Manufacturing shows slightly lower diversity ($4.70 \pm 0.46$) and engagingness ($4.90 \pm 0.30$), while Technology and Finance exhibit modest drops in coherence ($4.60 \pm 0.49$). These differences may reflect domain-specific communication patterns or the inherent complexity of certain topics. Overall, the results demonstrate that high-quality synthetic conversations can closely match real-world standards across multiple dimensions, especially when generated using structured, context-aware pipelines. However, achieving full diversity and engagingness remains challenging for baseline approaches, highlighting the importance of multi-turn, persona-driven synthesis for realistic team interactions.

## C.6 SYNTHETIC QUERY AND LABEL GENERATION

To benchmark contextual document generation, we synthesize realistic user queries and corresponding ground-truth labels using a multi-stage process grounded in the knowledge graph $\mathcal{G}$. This process ensures that each query is context-aware, temporally plausible, and paired with interpretable labels for evaluation.

**Step 1: Sampling User and Context.** We begin by sampling a target user node from $\mathcal{G}$, extracting the user's persona (role, tone, style, expertise) and their involvement in domains, topics, and phases. For each query, we randomly select either a phase or topic that the user is actively engaged in, ensuring diversity across document types (status report, email, FAQ).

**Step 2: Extracting Contextual Markers.** For the selected phase or topic, we traverse the graph to collect contextual markers from connected messages. These markers include:

- **Entities:** Projects, tools, concepts, people/roles.
- **Temporal Expressions:** Dates, deadlines, milestones.
- **User Actions:** Requests, suggestions, decisions.
- **Key Decisions:** Conclusions, owner assignments, approvals.
- **Unresolved Questions:** Blockers, concerns, open issues.
- **Mentioned Tools:** Software, platforms, systems.
- **Deliverable Sources:** URLs, file paths, document references.
- **Project Context:** Project name, topic, phase, status, owner, dates.

Markers are extracted using either LLM-based analysis or regex-based heuristics, with each marker linked to its source message for traceability.

**Step 3: Sampling Query Timestamp.** A realistic timestamp is sampled for each query, typically during or shortly after the relevant phase, to ensure temporal coherence. This timestamp is used to filter contextual markers and ground-truth messages, so that only information available up to the query time is considered.

**Step 4: Generating Intent Schema.** We construct a structured intent schema for each query, specifying:

- **Document Type:** Status report, email, or FAQ.
- **Target Audience:** Executives, team members, stakeholders, etc.
- **Temporal Scope:** Last week, past month, ongoing, upcoming, etc.
- **Detail Level:** Summary, detailed, comprehensive.
- **Tone:** Formal, technical, conversational, urgent, etc.
- **Visual Elements:** Charts, tables, dashboards, etc.
- **Format Instruction:** How to organize and present the document.
- **Document Structure:** Key sections to include.
- **Special Instruction:** Any specific requirements or constraints.

The intent schema is generated using an LLM prompt that incorporates the user profile and project context, ensuring that the schema is realistic and tailored to the scenario.

**Step 5: Synthesizing the User Query.** A natural language query is generated using the intent schema, contextual markers, and user persona. The LLM is prompted to produce a concise, context-aware request that reflects real workplace behavior, without explicitly mentioning document types. Example prompt:

Generate a concise but natural workplace request about fraud detection.
Context:
- Project: Treasury Modernization
- Topic: Fraud Detection
- User Role: Risk Analyst

The user needs information quickly, about recent progress, for leadership, including key progress
and what's coming up next.
Requirements:
1. Do not mention document types like "status report" or "email".
2. Use natural business language.
3. Make the request sound conversational and realistic.
4. Reference the project/topic naturally.
5. Subtly hint at the content areas needed without being too explicit.

**Example Output:**

Can you help me get up to speed on where we are with fraud detection—I need the key progress and
what's coming up next?

**Step 6: Ground-Truth Label Collection.** For each query, we collect the set of ground-truth messages from the graph that are relevant to the selected phase or topic and occurred before the query timestamp. These message IDs serve as the gold standard for context retrieval and evaluation.

**Step 7: Output Structure.** Each synthetic query is paired with rich metadata for evaluation:

- The generated query text.
- Document type, target node, user ID, timestamp.
- User persona and involvement context.
- Intent schema (structured label).
- Contextual markers (with source tracking).
- Ground-truth message IDs.

This pipeline produces a diverse set of synthetic queries and corresponding labels, enabling fine-grained evaluation of contextual document generation models. Each query is grounded in realistic user behavior, project context, and temporal constraints, with interpretable labels for intent, context, and ground-truth evidence.

### C.7 GOLDEN DOCUMENT GENERATION PIPELINE

To ensure the quality and reliability of reference documents for evaluation, we adopt an iterative optimization workflow that integrates LLM-based assessment, targeted editing, and manual curation. The following steps summarize the pipeline:

1. **Initial Draft Generation:**
   - For each synthetic query, generate an initial document draft using the best-performing LLM, grounded in the relevant context and intent schema.
2. **Section Chunking:**
   - Split the draft into logical sections (e.g., by markdown headings or dividers) for fine-grained evaluation.
3. **Automated Evaluation:**
   - For each section, apply a suite of LLM-as-judge metrics (accuracy, relevance, readability, redundancy, tone, detail level, specificity) to identify issues and assign scores.
4. **Targeted Editing:**
   - Sections with low scores or flagged issues are revised using targeted LLM editing prompts or manual intervention, focusing on factual grounding, evidence attribution, and persona fidelity.
   - Merge updated sections back into the document, preserving original order.
5. **Iterative Refinement:**
   - Repeat evaluation and editing until all sections meet predefined quality thresholds.
6. **Final Assembly:**
   - Combine all revised sections into the final golden document, which serves as the reference for benchmarking.

---

**Algorithm 1** Golden Document Generation Workflow

---

1: **for** each synthetic query **do**
2:    Generate initial draft using best-performing LLM (grounded in context and intent)
3:    Chunk draft into logical sections
4:    **repeat**
5:      Evaluate each section with LLM-as-judge metrics (accuracy, structure, personalization)
6:      Revise low-scoring/problematic sections via targeted LLM editing or manual curation
7:    **until** all sections meet quality thresholds
8:    Assemble revised sections into the final golden document
9: **end for**

---

**Pseudocode Illustration**

**Quality Control** Each golden document is validated for coverage of required sections, correct evidence IDs, and persona fidelity. The iterative workflow ensures that reference documents are both contextually accurate and personalized, supporting robust evaluation of model outputs.

## D  BENCHMARK DETAILS

### D.1  USER PROFILE INFERENCE PROMPTS AND EXAMPLE

**Prompt Template** The following prompt is used to infer the user profile from a set of messages:

> Analyze the following messages from user '{user_id}' and infer their professional profile.
>
> Messages: {message_content}
>
> Based on these messages, please infer their profile using ONLY the following predefined options:
>
> - Professional role: Choose the most fitting role from common workplace positions (e.g., Product Manager, Data Analyst, IT Systems Lead, Software Engineer, Project Manager, Business Analyst, etc.)
> - Expertise level: Choose from [novice, intermediate, expert]
> - Communication style: Choose from [concise, elaborative, standard, bullet-pointed]
> - Tone: Choose from [formal, professional, technical, conversational, direct, persuasive, empathetic, accessible]
> - Domain knowledge areas: List relevant technical/business domains
> - Project involvement/responsibilities: List inferred responsibilities
> - Confidence score (0-1) for your inference
>
> IMPORTANT: You must select exact values from the predefined options for expertise_level, communication_style, and tone. Do not use synonyms or variations.
>
> Respond in JSON format:
>
> ```
> {
>   "role": "...",
>   "expertise_level": "novice|intermediate|expert",
>   "communication_style": "concise|elaborative|standard|bullet-pointed",
>   "tone": "formal|professional|technical|conversational|direct|persuasive|empathetic
>   "domain_knowledge": ["...", "..."],
>   "project_involvement": ["...", "..."],
>   "confidence_score": 0.85
> }
> ```

**Example** Suppose the user messages are:

```
Message 1: "Let's ensure the Q3 financial summary is ready for the executive review."
Message 2: "Please use bullet points for key risks and a table for compliance status."
Message 3: "Our audience is the management team; keep the tone formal and concise."
```

**User Profile Output:**

```
{
  "role": "Finance Manager",
  "expertise_level": "expert",
  "communication_style": "concise",
  "tone": "formal",
  "domain_knowledge": ["finance", "compliance"],
  "project_involvement": ["executive reporting", "risk assessment"],
  "confidence_score": 0.95
}
```

## D.2 INTENT DETECTION PROMPTS AND EXAMPLE

**Prompt Template**   The following prompt is used for intent schema extraction:

Analyze the following user query and extract the structured intent for document generation.

User Query: "{query}"

Context Information: - Document Type: {context.get('document_type', 'Unknown')} - Contextual Markers: {context.get('contextual_markers', {})}

Extract and structure the following intent components using ONLY the predefined options:

- Document type: Choose from [status_report, email, faq]
- Target audience: Choose from [executives, team_members, stakeholders, management, clients, board]
- Temporal scope: Choose from [last_week, past_month, quarter, project_start, ongoing, upcoming, last_two_weeks]
- Detail level: Choose from [summary, detailed, comprehensive, high_level]
- Tone: Choose from [formal, technical, conversational, executive, urgent, celebratory, accessible]
- Format instruction: Describe specific formatting requirements (bullet_points, paragraphs, tables_charts, mixed, etc.)
- Document structure: List the main sections or topics that should be covered
- Visual elements: List any visual elements needed (charts_and_graphs, progress_bars, status_tables, etc.)

IMPORTANT: You must select exact values from the predefined options for document_type, target_audience, temporal_scope, detail_level, and tone. Do not use synonyms or variations.

Respond in JSON format:

```
{
  "document_type": "status_report|email|faq",
  "target_audience": "executives|team_members|stakeholders|management|clients|board",
  "temporal_scope": "last_week|past_month|quarter|project_start|ongoing|upcoming|last
  "detail_level": "summary|detailed|comprehensive|high_level",
  "tone": "formal|technical|conversational|executive|urgent|celebratory|accessible",
  "format_instruction": "...",
  "document_structure": ["...", "..."],
  "visual_elements": ["...", "..."]
}
```

**Example**   Suppose the user query is: `Generate a quarterly status report for management, focusing on compliance and financials, using bullet points and tables where appropriate.`

**Intent Schema Output:**

```
{
  "document_type": "status_report",
```

```
"target_audience": "management",
"temporal_scope": "quarter",
"detail_level": "detailed",
"tone": "formal",
"format_instruction": "bullet points for risks, tables for compliance and financials",
"document_structure": ["executive summary", "financial overview", "compliance status", "]
"visual_elements": ["tables"]
}
```

### D.3 Context Retrieval Prompt and Example

**Prompt Template**  The following prompt is used for context retrieval (see retrieve_relevant_context):

> Given a user query and document intent, select the most relevant messages from the conversation
> history.
> User Query: "{query}"
> Document Intent:
>   • - Document Type: {intent.document_type}
>   • Target Audience: {intent.target_audience}
>   • Temporal Scope: {intent.temporal_scope}
>   • Detail Level: {intent.detail_level}
>   • Tone: {intent.tone_preference}
>   • Specific Topics: {', '.join(intent.specific_topics) if intent.specific_topics else 'None'}
> Available Messages (all temporally filtered messages): {formatted messages}
> These are all {N} temporally filtered messages (messages that occurred before the query timestamp).
> Select the {num_target_messages} most relevant messages that would be needed to generate the
> requested document.
> Consider: 1. Temporal relevance (matches the temporal scope) 2. Content relevance (contains
> information needed for the document) 3. Author relevance (messages from key stakeholders) 4.
> Topic alignment (discusses relevant topics) 5. No duplicated or near-duplicate messages
> Respond with a JSON list of message IDs in order of relevance:
>
> ```
> ["Msg_101", "Msg_115", "Msg_120", ...]
> ```

**Example**  Suppose the available messages are:

```
[Msg_101] Alice (2025-07-01): "Q3 financials are finalized."
[Msg_102] Bob (2025-07-02): "Compliance review scheduled for July 10."
[Msg_103] Alice (2025-07-03): "Key risk: delayed vendor payments."
[Msg_104] Carol (2025-07-04): "Team lunch next Friday."
[Msg_105] Bob (2025-07-05): "All compliance documents uploaded."
```

**User Query:**  Generate a quarterly status report for management, focusing on compliance and financials.

**Intent Schema:**

```
{
  "document_type": "status_report",
  "target_audience": "management",
  "temporal_scope": "quarter",
  "detail_level": "detailed",
  "tone": "formal",
  "specific_topics": ["compliance", "financials"]
}
```

**Model Output (Relevant Message IDs):**

```
["Msg_101", "Msg_102", "Msg_103", "Msg_105"]
```

## D.4 REFERENCE-FREE LLM JUDGES

For reference-free evaluation in the PersonaContextWeaver benchmark, we employ an LLM-as-a-Judge protocol implemented in `document_generation.py`. This protocol uses a large language model to score generated documents across six key dimensions, using a systematic, step-by-step prompt and JSON output for consistency and transparency.

**Evaluation Dimensions**  Each generated document is evaluated across six dimensions, each scored on a scale from 1 to 5:

- **Personalization Fidelity**: Assesses how well the document reflects the intended user persona, including role, tone, audience, and temporal scope.
- **Factuality**: Measures the accuracy of claims made in the document, ensuring they are supported by cited evidence from the source context.
- **Citation Quality**: Evaluates whether citations are correctly formatted, relevant, appropriately placed, and sufficiently cover factual content.
- **Fluency**: Examines the clarity, grammatical correctness, and readability of the document, as well as its appropriateness for the target audience.
- **Structure**: Reviews the logical organization, formatting, and completeness of the document, including adherence to professional standards.
- **Temporal and Task Accuracy**: Checks whether the document content aligns with the specified timeframe and reflects the correct project phase or task context.

**Prompt Template**  The following prompt is programmatically constructed and sent to the LLM for each evaluation (see `evaluate_document_quality` in `document_generation.py`):

Evaluate the quality of the following generated document using a systematic evaluation process.

ORIGINAL USER QUERY: {query}

{intent context}{profile context}{temporal context}

DOCUMENT TO EVALUATE: {document.content}

CITATIONS USED: {citations_json}

EVALUATION PROCESS: Evaluate each metric systematically using the specific guidelines below:

FOR EACH METRIC, FOLLOW THESE DETAILED STEPS:

**1. Personalization Fidelity Evaluation**

- Identify document type from structure and content
- Compare identified type with expected type specification
- Analyze tone and style used throughout document
- Verify tone matches target audience and requirements
- Check temporal scope references in content
- Assess if detail level matches specified requirements
- Review format compliance with specified requirements
- Score 1–5: How well does document reflect intended specifications?

**2. Factuality Evaluation**

- Identify all factual claims and assertions in document
- For each claim, locate corresponding citation and source
- Verify facts against actual cited source content
- Check for any unsupported or speculative statements
- Look for contradictions between claims and sources
- Assess overall factual accuracy and evidence backing
- Score 1–5: How well are claims supported by evidence?

**3. Citation Quality Evaluation**

- Check all citation formats for proper [Msg_XXX] structure

- Verify each cited message ID exists and is accessible
- For each citation, confirm it supports the accompanying claim
- Assess appropriateness of citation placement in text
- Evaluate sufficiency of citation coverage for factual content
- Check for any missing citations for factual statements
- Score 1–5: How accurate and appropriate are the citations?

**4. Fluency Evaluation**

- Read through document checking for clarity and comprehension
- Identify any grammatical errors or awkward phrasing
- Assess logical flow and transitions between ideas
- Evaluate language appropriateness for target audience
- Check for engaging and professional writing style
- Review overall readability and coherence
- Score 1–5: How clear and well-written is the document?

**5. Structure Evaluation**

- Analyze overall document organization and logical flow
- Check if structure is appropriate for document type
- Evaluate headings, formatting, and visual layout
- Assess completeness of necessary sections
- Review adherence to professional document standards
- Check for logical progression from introduction to conclusion
- Score 1–5: How well-organized and structured is the document?

**6. Temporal and Task Accuracy Evaluation**

- Identify temporal scope specified in requirements
- Check all time references in document for accuracy
- Cross-reference content timeframe with citation timestamps
- Verify temporal expressions (dates, deadlines) are appropriate
- Assess if content reflects correct project phase/period
- Look for any temporal inconsistencies or anachronisms
- Score 1–5: How accurately does content align with specified timeframe?

FINAL SCORING: For each metric, provide a score (1–5) based on your systematic evaluation.

Respond in JSON format:

```
{
  "personalization_fidelity": 4,
  "factuality": 3,
  "citation_quality": 4,
  "fluency": 5,
  "structure": 4,
  "temporal_task_accuracy": 4,
  "overall_score": 4.0,
  "detailed_feedback": "METRIC-BY-METRIC EVALUATION: ... [OVERALL SUMMARY] ..."
}
```

**Implementation Details**

- **Model:** All evaluations are performed using GPT-4.1 or newer models.
- **Parameters:** Temperature is set to 0.1 for consistency. The prompt is delivered as a user message, and the LLM is instructed to respond only with valid JSON.
- **Scoring:** Scores for each dimension are aggregated into an overall metric. Qualitative feedback is provided for interpretability.
- **Automation:** The evaluation pipeline is fully automated and integrated in `document_generation.py`, ensuring reproducibility.

**Example Output**    The following is a representative JSON output from the LLM-as-a-Judge protocol:

```
{
   "personalization_fidelity": 4,
   "factuality": 4,
   "citation_quality": 4,
   "fluency": 5,
   "structure": 5,
   "temporal_task_accuracy": 5,
   "detailed_feedback": "METRIC-BY-METRIC EVALUATION: [PERSONALIZATION FIDELITY] Steps
       1a-1g assessment: The document is clearly a status_report, matching the expected
       type. The tone is executive and professional, suitable for management, and the
       detail level is comprehensive. The temporal scope is explicitly stated as Q3 2025
        and is referenced throughout. Formatting requirements are met: narrative
       paragraphs for summaries, bullet points for actions/issues, and tables for
       financials and compliance tracking. Minor deduction for placeholder financial
       figures (X, Y, Z, etc.), which slightly reduces personalization completeness. [
       FACTUALITY] Steps 2a-2f assessment: All major factual claims (milestone
       completion, compliance phase status, actions taken, risks, and open issues) are
       supported by the provided citations. There are no unsupported or speculative
       statements, and no contradictions with the cited sources. The only minor issue is
        the lack of actual financial data, but this is transparently noted as pending. [
       CITATION QUALITY] Steps 3a-3f assessment: All citations use the correct [Msg_XXX]
        format and refer to valid message IDs present in the citation list. Citations
       are appropriately placed after factual claims and cover all key assertions. There
        are no missing citations for factual statements. Placement is generally
       appropriate, though some sections could benefit from more granular citation (e.g
       ., individual bullet points). [FLUENCY] Steps 4a-4f assessment: The document is
       clear, well-written, and free of grammatical errors. The language is professional
        and appropriate for an executive audience. Transitions between sections are
       logical, and the writing style is engaging and concise. Readability and coherence
        are excellent. [STRUCTURE] Steps 5a-5f assessment: The document is well-
       organized, with all required sections present and clearly labeled. Headings,
       bullet points, and tables are used effectively, and the logical flow from
       introduction to conclusion is strong. The structure adheres to professional
       standards for status reports. [TEMPORAL ACCURACY] Steps 6a-6f assessment: The
       temporal scope (Q3 2025) is consistently referenced, and all content aligns with
       the specified period. Citation timestamps are within the correct timeframe, and
       there are no temporal inconsistencies or anachronisms. The document accurately
       reflects the current project phase and period. [OVERALL SUMMARY] Key strengths
       include strong alignment with specifications, comprehensive coverage of required
       topics, clear and professional writing, and robust structure. The main area for
       improvement is the use of placeholder financial data, which, while transparently
       noted, slightly reduces the document's completeness and personalization. Overall,
        the document is highly effective and meets the requirements for a management-
       level quarterly status report."
}
```

## D.5   DETAILS OF BENCHMARK RESULTS BY DATASETS

We provide detailed evaluation results on each dataset in Table 7 8, 9, 10.

## D.6   DETAILS OF INTENT DETECTION BY FIELDS

Table 11 shows the per-field mean Precision, Recall, and F1 scores for intent schema extraction across five LLMs (GPT-4o, GPT-4.1, O4-mini, GPT-5-chat, GPT-5). The benchmark task requires models to infer structured fields such as `Document Type`, `Target Audience`, `Temporal Scope`, `Detail Level`, and `Tone Preference` from realistic user queries.

First, all models achieve near-perfect performance on `Document Type` (F1 $\geq$ 0.94). This result highlights that explicit cues in user queries—such as requests for an "overview," "summary," or "update"—enable LLMs to reliably identify the intended document type. The consistently high scores across models suggest that this field is well-aligned with surface-level lexical patterns and is less sensitive to model architecture or prompt ambiguity.

Table 7: **Performance Metrics for Finance Dataset**

| Task | Metric | GPT-4o | GPT-4.1 | O4-mini | GPT-5-chat | GPT-5 |
|---|---|---|---|---|---|---|
| **Profile & Intent Detection** | User Profile Accuracy ↑ | 0.37 | 0.40 | 0.40 | 0.37 | 0.39 |
| | Intent Accuracy ↑ | 0.52 | 0.49 | 0.49 | 0.50 | 0.46 |
| **Context Filtering** | Precision ↑ | 0.30 | 0.17 | 0.18 | 0.19 | 0.23 |
| | Recall ↑ | 0.15 | 0.15 | 0.12 | 0.18 | 0.23 |
| | F1-score ↑ | 0.18 | 0.15 | 0.13 | 0.18 | 0.23 |
| **Document Generation (Reference-Free)** | Citation Accuracy (0.0 - 1.0) ↑ | 0.10 | 0.13 | 0.14 | 0.16 | 0.20 |
| | Personalization Fidelity (1-5) ↑ | 4.40 | 4.10 | 4.23 | 4.80 | 4.25 |
| | Factuality (1-5) ↑ | 4.23 | 3.80 | 4.05 | 4.98 | 3.38 |
| | Fluency (1-5) ↑ | 4.98 | 5.00 | 4.95 | 5.00 | 4.95 |
| | Structure (1-5) ↑ | 4.50 | 4.50 | 4.85 | 4.95 | 4.75 |
| | Temporal Accuracy (1-5) ↑ | 4.28 | 4.20 | 4.60 | 4.93 | 4.03 |
| **Document Generation (Reference-based)** | ROUGE-1 ↑ | 0.35 | 0.38 | 0.29 | 0.32 | 0.40 |
| | ROUGE-L ↑ | 0.13 | 0.14 | 0.11 | 0.12 | 0.13 |
| | METEOR ↑ | 0.22 | 0.23 | 0.14 | 0.19 | 0.24 |

Table 8: **Performance Metrics for Healthcare Dataset**

| Task | Metric | GPT-4o | GPT-4.1 | O4-mini | GPT-5-chat | GPT-5 |
|---|---|---|---|---|---|---|
| **Profile & Intent Detection** | User Profile Accuracy ↑ | 0.43 | 0.43 | 0.40 | 0.41 | 0.42 |
| | Intent Accuracy ↑ | 0.53 | 0.47 | 0.47 | 0.48 | 0.43 |
| **Context Filtering** | Precision ↑ | 0.25 | 0.21 | 0.22 | 0.24 | 0.26 |
| | Recall ↑ | 0.15 | 0.19 | 0.17 | 0.22 | 0.26 |
| | F1-score ↑ | 0.16 | 0.20 | 0.18 | 0.22 | 0.26 |
| **Document Generation (Reference-Free)** | Citation Accuracy (0.0 - 1.0) ↑ | 0.08 | 0.14 | 0.15 | 0.17 | 0.24 |
| | Personalization Fidelity (1-5) ↑ | 4.38 | 4.13 | 4.60 | 4.73 | 4.35 |
| | Factuality (1-5) ↑ | 4.30 | 3.75 | 4.43 | 4.98 | 3.50 |
| | Fluency (1-5) ↑ | 4.98 | 5.00 | 4.95 | 5.00 | 4.98 |
| | Structure (1-5) ↑ | 4.53 | 4.55 | 4.98 | 4.95 | 4.75 |
| | Temporal Accuracy (1-5) ↑ | 4.25 | 4.18 | 4.73 | 4.95 | 4.13 |
| **Document Generation (Reference-based)** | ROUGE-1 ↑ | 0.35 | 0.37 | 0.29 | 0.31 | 0.41 |
| | ROUGE-L ↑ | 0.13 | 0.13 | 0.11 | 0.12 | 0.14 |
| | METEOR ↑ | 0.22 | 0.21 | 0.14 | 0.17 | 0.23 |

Second, moderate scores are observed for `Target Audience` (F1 $\approx$ 0.60-0.70) and `Detail Level`. For `Target Audience`, models benefit from queries that mention specific roles or stakeholders (e.g., "management," "leadership," "stakeholders"), but performance drops when the audience is implied rather than explicit. For `Detail Level`, GPT-4o leads (F1 = 0.50), indicating some ability to distinguish between requests for summaries versus detailed breakdowns. However, the variability across models suggests that granularity cues are often subtle and may require more sophisticated context modeling.

Third, `Temporal Scope` and `Tone Preference` remain challenging for all models (F1 $\leq$ 0.36 and F1 $\leq$ 0.17, respectively). Temporal expressions such as "recent progress," "so far," or "on the horizon" are frequently present in queries, but models struggle to consistently map these to a structured temporal field. Similarly, tone is often implicit—embedded in the phrasing or urgency of the request—making it difficult for models to extract reliably. These results indicate that fields requiring deeper semantic or pragmatic reasoning are not yet robustly handled by current LLMs.

Fourth, GPT-4o and GPT-4.1 demonstrate balanced performance across most fields, suggesting that effective generalization and instruction following capabilities. In contrast, GPT-5 variants underperform in extracting detail level and tone, possibly reflecting differences in model tuning, context window management, or training data emphasis. These findings underscore the importance of aligning model selection and prompt design with the specific personalization and grounding requirements of downstream applications, especially for tasks involving nuanced or implicit user intent.

In summary, these results reveal that while LLMs are highly effective at extracting explicit and well-cued fields, substantial challenges remain for fields that require nuanced contextual or stylistic reasoning. To advance intent schema extraction, future work should focus on enhancing models' ability to interpret implicit cues, leverage richer context representations, and incorporate prompt engineering strategies that clarify temporal and tonal requirements. Additionally, expanding training data to include more diverse examples of implicit intent and developing evaluation

Table 9: **Performance Metrics for Manufacturing Dataset**

| Task | Metric | GPT-4o | GPT-4.1 | O4-mini | GPT-5-chat | GPT-5 |
|---|---|---|---|---|---|---|
| **Profile & Intent Detection** | User Profile Accuracy ↑ | 0.58 | 0.60 | 0.56 | 0.54 | 0.48 |
| | Intent Accuracy ↑ | 0.53 | 0.49 | 0.48 | 0.52 | 0.46 |
| **Context Filtering** | Precision ↑ | 0.32 | 0.17 | 0.10 | 0.20 | 0.25 |
| | Recall ↑ | 0.14 | 0.14 | 0.09 | 0.17 | 0.25 |
| | F1-score ↑ | 0.17 | 0.15 | 0.09 | 0.18 | 0.25 |
| **Document Generation (Reference-Free)** | Citation Accuracy (0.0 - 1.0) ↑ | 0.11 | 0.11 | 0.10 | 0.16 | 0.22 |
| | Personalization Fidelity (1-5) ↑ | 4.43 | 4.10 | 4.53 | 4.60 | 4.19 |
| | Factuality (1-5) ↑ | 4.35 | 3.48 | 4.03 | 4.88 | 3.48 |
| | Fluency (1-5) ↑ | 5.00 | 5.00 | 4.88 | 4.88 | 4.93 |
| | Structure (1-5) ↑ | 4.50 | 4.35 | 4.83 | 4.83 | 4.72 |
| | Temporal Accuracy (1-5) ↑ | 4.33 | 4.18 | 4.78 | 4.78 | 4.18 |
| **Document Generation (Reference-based)** | ROUGE-1 ↑ | 0.35 | 0.38 | 0.27 | 0.31 | 0.42 |
| | ROUGE-L ↑ | 0.13 | 0.13 | 0.10 | 0.12 | 0.13 |
| | METEOR ↑ | 0.24 | 0.23 | 0.13 | 0.18 | 0.25 |

Table 10: **Performance Metrics for Technology Dataset**

| Task | Metric | GPT-4o | GPT-4.1 | O4-mini | GPT-5-chat | GPT-5 |
|---|---|---|---|---|---|---|
| **Profile & Intent Detection** | User Profile Accuracy ↑ | 0.54 | 0.56 | 0.45 | 0.49 | 0.48 |
| | Intent Accuracy ↑ | 0.55 | 0.56 | 0.53 | 0.51 | 0.46 |
| **Context Filtering** | Precision ↑ | 0.21 | 0.22 | 0.13 | 0.22 | 0.25 |
| | Recall ↑ | 0.12 | 0.16 | 0.09 | 0.19 | 0.25 |
| | F1-score ↑ | 0.14 | 0.17 | 0.10 | 0.20 | 0.25 |
| **Document Generation (Reference-Free)** | Citation Accuracy (0.0 - 1.0) ↑ | 0.12 | 0.15 | 0.11 | 0.19 | 0.22 |
| | Personalization Fidelity (1-5) ↑ | 4.35 | 4.20 | 4.31 | 4.83 | 4.19 |
| | Factuality (1-5) ↑ | 4.35 | 3.70 | 4.30 | 4.98 | 3.48 |
| | Fluency (1-5) ↑ | 5.00 | 5.00 | 4.93 | 5.00 | 4.93 |
| | Structure (1-5) ↑ | 4.50 | 4.43 | 4.90 | 4.93 | 4.72 |
| | Temporal Accuracy (1-5) ↑ | 4.33 | 4.23 | 4.49 | 4.90 | 4.18 |
| **Document Generation (Reference-based)** | ROUGE-1 ↑ | 0.35 | 0.37 | 0.28 | 0.31 | 0.40 |
| | ROUGE-L ↑ | 0.13 | 0.13 | 0.11 | 0.12 | 0.13 |
| | METEOR ↑ | 0.23 | 0.22 | 0.13 | 0.19 | 0.24 |

protocols that reward semantic and pragmatic understanding will be critical for improving performance on complex, real-world document generation tasks.

Table 11: Per-field mean Precision / Recall / F1 for Intent Schema Extraction Across Models

| Field | GPT-4o | GPT-4.1 | O4-mini | GPT-5-chat | GPT-5 |
|---|---|---|---|---|---|
| Target Audience | 0.70 / 0.70 / 0.70 | 0.66 / 0.66 / 0.66 | 0.62 / 0.62 / 0.62 | 0.64 / 0.64 / 0.64 | 0.60 / 0.60 / 0.60 |
| Temporal Scope | 0.30 / 0.30 / 0.30 | 0.33 / 0.33 / 0.33 | 0.35 / 0.35 / 0.35 | 0.34 / 0.34 / 0.34 | 0.36 / 0.36 / 0.36 |
| Detail Level | 0.50 / 0.50 / 0.50 | 0.36 / 0.36 / 0.36 | 0.43 / 0.43 / 0.43 | 0.37 / 0.37 / 0.37 | 0.29 / 0.29 / 0.29 |
| Document Type | 0.99 / 0.99 / 0.99 | 1.00 / 1.00 / 1.00 | 0.94 / 0.94 / 0.94 | 0.99 / 0.99 / 0.99 | 0.98 / 0.98 / 0.98 |
| Tone Preference | 0.16 / 0.16 / 0.16 | 0.17 / 0.17 / 0.17 | 0.12 / 0.12 / 0.12 | 0.15 / 0.15 / 0.15 | 0.08 / 0.08 / 0.08 |

