# OpenReview forum: "Shared Contexts, Personalized Outputs: A Benchmark for Document Generation"
_ICLR.cc/2026/Conference — ICLR 2026 Conference Withdrawn Submission_

### Official Review · Reviewer_CENQ · 2025-10-21

**Soundness:** 1
**Presentation:** 2
**Contribution:** 2
**Rating:** 0
**Confidence:** 4

**Summary:**

This paper proposes a new benchmark, PersonaContextWeaver, for evaluating the performance of large language models (LLMs) in the Personalized Contextual Document Generation (PCDG) task. This framework constructs multi-user, multi-domain conversational contexts through a knowledge-graph-based synthetic pipeline. It defines three subtasks (personalized intent detection, context filtering, and document generation) and provides a multi-dimensional evaluation protocol to measure model performance in terms of personalization, contextual consistency, and reasoning. The authors conducted experiments on several state-of-the-art LLMs (GPT-4 and GPT-5 series) and found that existing models still have significant shortcomings in terms of personalized generation and factual consistency.

**Strengths:**

1. This paper focuses on personalized document generation (PCDG), a field that combines long-context reasoning and user modeling and is a less systematically explored area in LLM research.
2. PersonaContextWeaver decomposes the complex generation task into three interpretable subtasks (intent detection, context filtering, and document generation). This pipeline-based evaluation design helps researchers analyze performance bottlenecks in different stages of the model.
3. The authors employ a knowledge-graph-driven synthesis pipeline to generate controllable multi-user conversational data and introduce personalized variables (role, tone, and expertise level).
4. The authors systematically compare the performance of various mainstream LLMs and conduct analysis using both reference-based and reference-free evaluation criteria, revealing the limitations of traditional metrics (such as ROUGE) for diverse personalization tasks.

**Weaknesses:**

1. The paper claims the generated multi-user conversations are "realistic and diverse," but relies solely on LLM evaluation (G-Eval) as a measure of authenticity, lacking manual verification or quantitative comparison with real conversational data. Furthermore, the authors fail to demonstrate how changing a user's persona attributes (e.g., changing their style from Polite to Rude) produces stable and predictable changes in the generated conversations. The paper simply describes the inclusion of these attributes in prompts, but lacks the necessary ablation studies or correlation analysis to demonstrate that this manipulation is effective and reliable. This is a core claim repeatedly emphasized in the abstract and introduction, but the evidence is currently incomplete, raising questions about whether the dataset reflects real user behavior.
2. The paper provides multiple metrics (e.g., personalization fidelity, factuality, and temporal accuracy), but lacks analysis of inter-rater reliability and stability, nor does it provide a weighting basis or error range for each metric, limiting the credibility of these claims. Furthermore, the paper mentions automatically extracting relevant message IDs from the knowledge graph as ground truth for the context filtering task. However, the paper's description of this extraction process is overly vague, failing to clearly define how relevance is determined from the graph structure.
3. In conclusion, the authors emphasize that GPT-5-chat achieves the best personalization fidelity and factuality. However, in Table 6, GPT-5 achieves the highest citation accuracy but significantly lower factuality. This inconsistency is not clearly explained.
4. PersonaContextWeaver comprises multiple complex modules (graph synthesis, task decomposition, and evaluation protocol), but the authors fail to demonstrate the independent contribution of each module to the final performance or interpretability, making it difficult to fully validate their claimed innovations.
5. The experiments evaluate only a few models from the OpenAI family and lack testing of other leading open-source models (such as the Llama family and Qwen), limiting the generalizability of their conclusions. More seriously, the paper uses GPT-4.1 as the LLM-as-judge to evaluate all models, including other models in the GPT family. LLM-as-judge is known to suffer from self-preference bias, meaning that a model may tend to give high scores to models in the same family. The authors fail to mention or discuss this potential bias in the paper, which undermines their conclusion that GPT-5-chat is the best-performing model.

**Questions:**

1. Please explain how the control variables (e.g., topic diversity and persona overlap) in the synthetic conversation generation were quantified or validated? Is there a risk of overfitting to the model's specific prompt patterns?
2. In the evaluation framework, the reference-free portion uses LLM-as-judge evaluation. Please provide scoring consistency (e.g., variance under different judge models or different seeds).
3. In Table 6, GPT-5-chat and GPT-5 show opposite trends in the factuality and citation accuracy metrics. Please explain the reasons for this discrepancy.
4. In the multi-dimensional metrics, how are the scores of different dimensions weighted and aggregated? Were significance tests performed to support the claims?

---

### Official Review · Reviewer_CPyG · 2025-10-30

**Soundness:** 3
**Presentation:** 3
**Contribution:** 2
**Rating:** 2
**Confidence:** 3

**Summary:**

This paper introduces PersonaContextWeaver, a new benchmark for evaluating large language models (LLMs) on personalized contextual document generation (PCDG) tasks. Unlike existing long-context or persona-based datasets, this framework simulates multi-user, multi-stage collaborative dialogues through graph-based synthetic data generation, where each user shares a common context but has unique personas and document needs (e.g., reports, emails, FAQs).

The benchmark decomposes PCDG into three interpretable subtasks: personalized intent detection, context filtering, and document generation, each with explicit gold annotations. It also proposes multi-dimensional evaluation metrics, combining reference-based (ROUGE, METEOR) and reference-free (LLM-as-judge) assessments covering personalization, factuality, structure, and temporal coherence. Experiments across five OpenAI models (GPT-4.1, GPT-4o, GPT-5, GPT-5-chat, O4-mini) show varying strengths: GPT-5-chat achieves the best personalization and factual alignment, revealing how model architectures affect nuanced user adaptation.

**Strengths:**

Novel Benchmark Design: The first to explicitly couple shared context and personalized document generation, filling a clear gap between long-context understanding and persona-grounded output generation.

Graph-based Synthetic Data: Ingenious use of knowledge-graph traversal to create rich, controllable, multi-user dialogue data that remains coherent and realistic.

Interpretable Pipeline: The three-stage decomposition (intent → filtering → generation) allows diagnostic evaluation — an important methodological contribution for understanding LLM reasoning steps.

Comprehensive Evaluation: The integration of reference-free “LLM-as-judge” metrics beyond ROUGE provides more human-like assessment of personalization and coherence.

Empirical Insights: Cross-model analysis highlights strengths and weaknesses of current LLMs, offering actionable insights for both research and deployment.

**Weaknesses:**

Synthetic Data Limitations: Despite its realism, the synthetic dialogues may lack the spontaneity and implicit cues of real multi-party communication.

Lack of Human Evaluation: No large-scale user study validates whether outputs truly meet human expectations of personalization.

Restricted Domain Coverage: Only four professional domains (tech, healthcare, manufacturing, finance); limited diversity in communication styles or cultural contexts.

Limited Baselines: Benchmark comparisons are confined to OpenAI models; lacks open-weight baselines (e.g., Claude, Gemini, Llama, Mistral).

Overreliance on LLM-as-judge: Potential circularity — evaluating LLMs using another LLM without calibration against human judgment.

Missing HCI Perspective: The paper focuses on benchmarking but doesn’t analyze user interaction patterns or how personalization impacts usability.

**Questions:**

How well does the benchmark generalize to real-world, noisy meeting data or enterprise workflows?

Can human-in-the-loop correction (e.g., user feedback loops) be integrated into the benchmark for adaptive evaluation?

Would RAG models benefit differently from the context-filtering stage?

How are biases in persona construction mitigated, especially when roles encode social or occupational stereotypes?

Could the authors release few-shot or incremental adaptation variants (to test continual personalization)?

**some suggested references:**

A Persona-Aware LLM-Enhanced Framework for Multi-Session Personalized Dialogue Generation.

Meetalk: Retrieval-Augmented and Adaptively Personalized Meeting Summarization with Knowledge Learning from User Corrections.

---

### Official Review · Reviewer_mDUM · 2025-10-31

**Soundness:** 3
**Presentation:** 3
**Contribution:** 2
**Rating:** 4
**Confidence:** 3

**Summary:**

This paper presents PersonaContextWeaver, a benchmark for Personalized Contextual Document Generation (PCDG), which evaluates how large language models generate user-tailored documents from shared, multi-user contexts. The framework synthesizes realistic conversations through a graph-based generation pipeline, decomposes the task into intent detection, context filtering, and document generation, and assesses models with both reference-based and LLM-as-judge metrics. Experiments across multiple OpenAI models (GPT-4.1, GPT-4o, GPT-5-chat, GPT-5) show strong fluency and coherence but persistent weaknesses in personalization and factual grounding. Overall, the benchmark is technically solid and reproducible, yet its contribution lies mainly in system design and evaluation rather than novel modeling or theoretical insight.

**Strengths:**

**1. Benchmark is well scoped and practically motivated**

The paper defines a clear and concrete problem of generating personalized documents from shared multi-user contexts, and builds a benchmark that directly evaluates this ability, which is not covered by existing long-context or summarization datasets.

**2. Implementation is carefully executed and well documented**

The benchmark pipeline, task decomposition, and evaluation scripts are transparent and reproducible, with detailed prompts and metrics that make it easy for others to replicate and extend.

**3. Empirical results provide informative baselines for future work**

The evaluation across GPT-4 and GPT-5 variants reveals current strengths and weaknesses in personalization and factual grounding, offering useful baselines for subsequent research on personalized text generation.

**Weaknesses:**

**1. Limited methodological novelty**

The contribution is mainly a benchmark, not a new modeling or learning approach. The design largely follows prior frameworks such as LongGenBench or PersonaBench, and the paper does not offer a distinct algorithmic insight or methodological innovation beyond task decomposition and evaluation setup.

**2. Reliance on synthetic data**

All conversational contexts and user queries are generated by LLMs rather than collected from real interactions. Although the authors report high “naturalness” and “coherence” scores, this self-generated setting may overestimate performance and does not confirm generalization to real, noisy, or domain-specific scenarios.

**3. Shallow analysis of results**

The results are mainly descriptive, summarizing metric differences without interpreting why models behave differently. For example, the observed trade-off between personalization and factuality (GPT-5-chat vs GPT-5) is not analyzed in relation to context filtering accuracy or grounding errors, leaving the reader uncertain about underlying causes.

**4. Evaluation circularity**

The benchmark heavily depends on LLM-as-judge scores, but no human evaluation or correlation check is presented. Without verifying whether these automated judgments align with human assessments, the credibility of reported metrics and model comparisons remains uncertain.

**Questions:**

See the weaknesses.

---

### Official Review · Reviewer_Xg5f · 2025-10-31

**Soundness:** 2
**Presentation:** 2
**Contribution:** 3
**Rating:** 2
**Confidence:** 3

**Summary:**

This paper presents PersonaContextWeaver, a benchmark for evaluating personalized contextual document generation. This benchmark decomposes the PCDG in interpretable sub-tasks and uses a multi-dimensional evaluation protocol to evaluate LLM’s capability.

**Strengths:**

The author’s approach of decomposing the personalized contextual document generation task into several sub-tasks and evaluating it from multiple dimensions is commendable. The domains and scenarios covered by the benchmark are also well-aligned with real-world situations.

**Weaknesses:**

1.	The authors used the ICLR 2025 paper template for writing and submission, but it is evident that they adjusted the margins to include more content, which violates the submission guidelines.
2.	The paper devotes the majority of its length (8 out of 9 pages) to describing the benchmark’s composition and evaluation methods, while only allocating one page to experimental results and analysis. This is highly unbalanced; the authors are advised to restructure the paper and include more intuitive case studies.

**Questions:**

Using an LLM-as-judge for evaluation can easily lead to score fluctuations due to the inherent performance variability of the judge model itself. Has the author considered training and open-sourcing a dedicated judge model to ensure consistency in evaluation results when different users employ this benchmark?

---

### Note · Authors · 2025-11-12

I have read and agree with the venue's withdrawal policy on behalf of myself and my co-authors.